# STEP-BACK PROFILING: Distilling User Interactions for Personalized Scientific Writing

## Abstract

Large language models (LLMs) excel at a variety of natural language processing tasks, yet they struggle to generate personalized content for individuals, particularly in real-world settings like scientific writing. Addressing this challenge, we introduce STEP-BACK PROFILING that personalizes LLMs by abstracting user interactions into concise profiles. Our approach effectively condenses user interaction history, distilling it into profiles that encapsulate essential traits and preferences of users, thus facilitating personalization that is both effective and user-specific. Importantly, STEP-BACK PROFILING is a low-cost and easy-to-implement technique that does not require additional fine-tuning. Through evaluation of the LaMP benchmark, which encompasses a spectrum of language tasks requiring personalization, our approach outperformed the baseline, showing improvements of up to 3.6 points. We curated the Personalized Scientific Writing (PSW) dataset to further study multi-user personalization in challenging real-world scenarios. This dataset requires the models to write scientific papers given specialized author groups with diverse academic backgrounds. On PSW, we demonstrate the value of capturing collective user characteristics via STEP-BACK PROFILING for collaborative writing. Extensive experiments and analysis validate our method's state-of-the-art performance and broader applicability – an advance that paves the way for more user-tailored scientific applications with LLMs.

## 1 Introduction

In recent years, Large Language Models (LLMs) have made significant strides in natural language understanding and generation, demonstrating human-parity performance on a wide range of tasks [Wei *et al.*, 2022b,a; Chowdhery *et al.*, 2023; OpenAI, 2023]. Moreover, the advent of LLM-driven language agents has revolutionized a myriad of user-facing applications, marking a game-changing breakthrough in the general AI capacity [Zhou *et al.*, 2023; Zhang *et al.*, 2023b; Shinn *et al.*, 2023; Qin *et al.*, 2023; Qiao *et al.*, 2024]. Con-

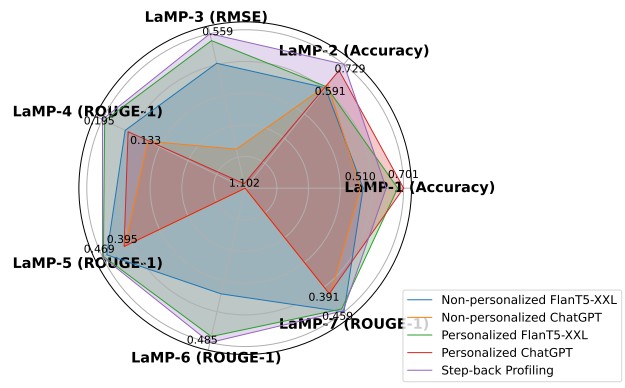

Figure 1: STEP-BACK PROFILING consistently improves the downstream task accuracy on the LaMP dataset.

currently, integrating LLMs with personalization paradigms has paved the way for a vast frontier in improving user-centric services and applications [Salemi *et al.*, 2023; Chen *et al.*, 2023; Zhiyuli *et al.*, 2023], as they provide a deeper understanding of users' accurate demands and interests than abstract vector-based information representations. By learning to characterize and emulate user-specific language patterns, personalized LLMs can enable more engaging and valuable interactions in domains such as dialogue [Wang *et al.*, 2019; Zhang *et al.*, 2019b; Character.AI, 2022], recommendation [Zhiyuli *et al.*, 2023; Wang *et al.*, 2023], role-playing [Shao *et al.*, 2023; Jiang *et al.*, 2023] and content creation [Cao *et al.*, 2023; Wei *et al.*, 2022c].

Prior work on personalizing language models [Salemi *et al.*, 2023; Tan and Jiang, 2023; Zhang *et al.*, 2023a; Chen *et al.*, 2023; Zhiyuli *et al.*, 2023] has shown promise, but primarily focused on learning user representations in a single-user context. For example, the LaMP benchmark [Salemi *et al.*, 2023] evaluates personalization given a single target user's historical interactions on tasks like citation prediction and product review generation. However, many real-world applications involve multiple users collaborating on a shared task, such as team-authored scientific papers.

Another practical challenge for LLM personalization is scaling to extensive user histories while respecting context

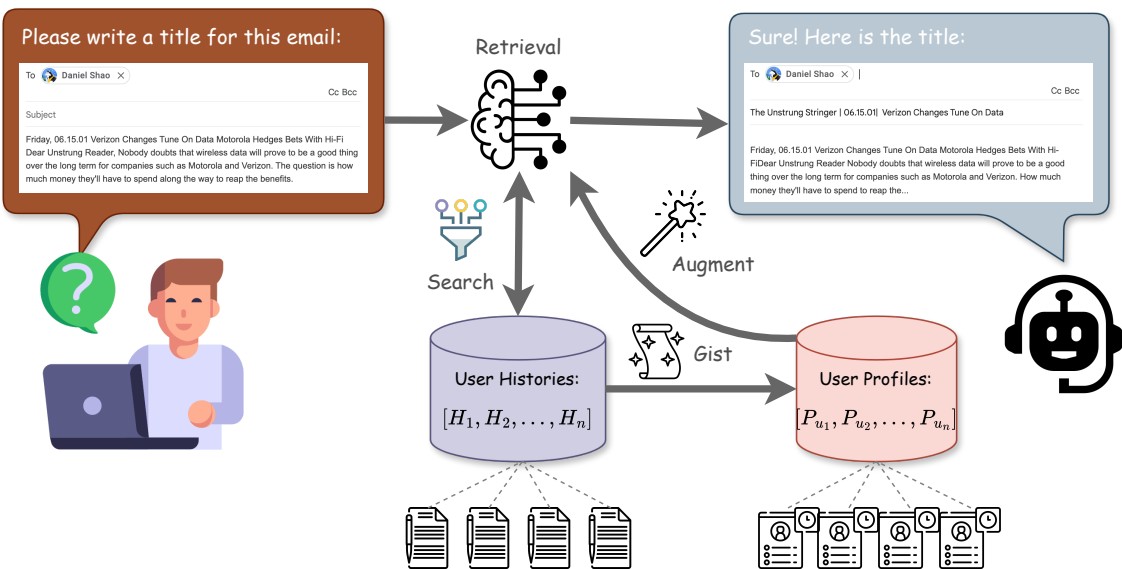

Figure 2: Overview of the STEP-BACK PROFILING Methodology. It applies the 'gist' abstraction function to the history of users and generates personalized output through an enhanced retrieval-augmented language model.

length limits [Shi *et al.*, 2023; Liu *et al.*, 2024]. Directly conditioning on raw personal histories quickly becomes infeasible as user data grows. Prior methods mostly use uncompressed history for personalization [Salemi *et al.*, 2023], which restricts the amount of user-specific information the model can utilize. This limits knowledge-intensive applications like scientific writing, where relevant information may be dilated across many documents.

This work proposes a training-free LLM personalization framework that addresses these challenges through STEP-BACK PROFILING – inspired by the ideas of gist memory [Lee *et al.*, 2024] and STEP-BACK PROMPTING [Zheng *et al.*, 2023] for information compression and abstraction, we distill individual user histories into concise profile representations that capture high-level concepts and language traits. This enables efficient memory management and allows the model to focus on salient user characteristics, grounding personalized generation without excess computation or laborious data collection [Chen *et al.*, 2023]. STEP-BACK PROFILING is a low-cost and easy-to-implement technique that operates directly on the pre-trained LLM without additional training. It can also complement parameter-efficient techniques of LLM finetuning [Hu *et al.*, 2021; Dettmers *et al.*, 2024; Sheng *et al.*, 2023]. We show that STEP-BACK PROFILING improves performance over standard personalization methods on the LaMP benchmark.

Moreover, we introduce the Personalized Scientific Writing (PSW) dataset to study multi-user personalization. PSW contains research papers collaboratively written by expert teams, and each author's background publications are used to construct profiles. Modeling a group's collective expertise is crucial for this task, as different paper sections may reflect knowledge associated with particular authors. PSW thus poses a challenging and realistic testbed for multi-user personalization, requiring both abstractions of individual exper-

tise and dynamic integration of diverse user traits throughout the collaborative writing process.

To summarize the contributions of this work:

1. A training-free STEP-BACK PROFILING approach that enables efficient and expressive personalization by abstracting user histories into trait-centric representations.

2. The Personalized Scientific Writing (PSW) dataset, a real-world benchmark for studying multi-user personalization with a novel task of collaborative expert writing.

3. A state-of-the-art performance of STEP-BACK PROFILING for single and multi-user personalization on diverse tasks in the LaMP and PSW benchmark without additional training.

## 2 STEP-BACK PROFILING

### 2.1 Motivation

Existing methods for personalizing language models struggle to effectively utilize user histories, particularly in the presence of extraneous details that can obscure the most pertinent information for a given task [Shi *et al.*, 2023; Liu *et al.*, 2024]. This challenge is magnified in multi-user scenarios, where models must efficiently extract and integrate knowledge from multiple users' histories. While retrieval-augmented methods, such as those employed in the LaMP benchmark [Salemi *et al.*, 2023], have made progress in scaling to more extensive user histories, they still operate on raw user data containing relevant and irrelevant details. To address these limitations, we introduce a STEP-BACK PROFILING approach that distills a user's raw history into a concise representation focusing on 'gist' representations and preferences, drawing inspiration from the STEP-BACK PROMPTING technique [Zheng *et al.*, 2023] and the READAGENT framework [Lee *et al.*, 2024]. Our approach aims to enable more efficient and effective personalization across diverse single and multi-user scenarios by

reasoning about higher-level traits instead of verbatim user history.

## 2.2 Procedure

Consider a set of $n$ users denoted by $U = \{u_1, u_2, \cdots, u_n\}$, where each user $u_i$ has a preference history $H_i = \{(x_{i1}, y_{i1}), (x_{i2}, y_{i2}), \cdots, (x_{im}, y_{im})\}$ consisting of $m$ input-output pairs. To effectively generate $P(y|x, H_U)$ based on users' preference history, we create a set of user profiles $P_U = \{P_{u_1}, P_{u_2}, \cdots, P_{u_n}\}$ using STEP-BACK PROFILING. The complete procedure involves the following steps:

**User Profile Gisting:** Each user's history is condensed into a short "gist" representation using an abstraction function $\text{Gist}(\cdot)$: $P_{u_i} = \text{Gist}(H_i)$. The "gist" captures the user's high-level traits and interests.

**Multi-User Profile Concatenation:** Individual user profiles $\{P_{u_1}, P_{u_2}, \cdots, P_{u_n}\}$ are concatenated to form a unified representation $P_U$: $P_U = [P_{u_1}; P_{u_2}; \cdots; P_{u_n}]$, where $[\cdot; \cdot]$ is a permutation-sensitive function combining the user profiles.

**Retrieval-Augmented Generation (Optional):** Relevant snippets from user histories $H_U$ may be retrieved for input $x$ using a retrieval function $\text{Retrieve}(\cdot)$: $R_i = \text{Retrieve}(x, H_i, k)$, where $R_i$ is a set of top-$k$ retrieved input-output snippets from user $u_i$'s history $H_i$. The retrieved snippets $R = \{R_1, R_2, \cdots, R_n\}$ can be concatenated with $x$ to form an augmented input $\hat{x}$: $\hat{x} = [x; R_1; R_2; \cdots; R_n]$.

**Personalized Output Generation:** The personalized language model generates an output $y$ by conditioning on the augmented input $\hat{x}$ (if retrieval is used) or the original input $x$, along with the concatenated user profile $P_U$: $y = \text{Generate}(\hat{x}, P_U)$. The generated output $y$ aligns with the user preferences captured by the STEP-BACK PROFILING while following the input $x$.

## 3 The Personalized Scientific Writing (PSW) Benchmark

We have extended the LaMP benchmark, introduced by Salemi *et al.* [2023], to evaluate multi-user scenarios. Our PSW benchmark includes four tasks, and we outline the data collection process for each task.

### 3.1 Problem Formulation

Personalized language models aim to generate outputs that follow a given input and align with the users' styles, preferences, and expertise. In multi-author collaborative writing, the Personalized Writing Styles (PSW) benchmark provides a framework for evaluating such models.

Each data entry in the PSW benchmark consists of four key components:

1. An input sequence $x$ serves as the model's input.

2. A target output $y$ that the model is expected to generate.

3. A set of user histories $H_U = \{H_{u_1}, H_{u_2}, \cdots, H_{u_k}\}$, where $k$ is the number of collaborating authors, and each entry $H_{u_i}$ contains historical input-output pairs for user $u_i$.

4. A set of author roles $C = \{c_1, c_2, \cdots, c_k\}$, each representing the role of the corresponding author $u_i$ in the collaborative writing process.

A personalized language model aims to generate an output $y$ that aligns with the conditional probability distribution $P(y|x, H_U, C)$. This means the model should produce an output that follows the input $x$ and the collaborating authors' writing styles, preferences, and expertise, as captured by their user histories $H_U$ and roles $C$.

By conditioning the language model's output on these additional factors, the PSW benchmark allows for the evaluation of personalized models that can adapt to the unique characteristics of individual authors in a collaborative writing environment.

### 3.2 Task Descriptions

**UP-0: Research Interest Generation:** Before all the PSW tasks, we create a benchmark for user profiling. This involves compiling a list of research interests that accurately reflect each author's expertise and research focus based on their publication history. To acquire the necessary information, we extract the research interests of each author from Google Scholar[1] by searching their name. Once we have this information, we incorporate it into the author's profile.

**PSW-1: Research Topic Generation:** This task aims to generate a list of research topics that capture the collaborating authors' joint expertise and research focus, given their user profiles. The generated research topics should be relevant to the authors' past publications and help identify potential research directions for their collaborative work. We use OpenAI's *gpt-4* model to automatically extract research topics from selected papers. The extracted topics are then linked to their respective papers and author profiles.

**PSW-2: Research Question Generation:** This task focuses on generating a set of research questions that align with the expertise and interests of the collaborating authors and are relevant to the target paper. The generated research questions should help guide the content and structure of the collaborative writing process. We automatically use OpenAI's *gpt-4* model to extract research questions from the selected papers for this task. The extracted research questions are then linked to their papers and author profiles.

**PSW-3: Paper Abstract Generation:** This task involves generating a paper abstract that summarizes the key points and contributions of the collaborative research paper, given the user profiles, research interests, target paper title, and research questions. The generated abstract should incorporate the writing styles and preferences of the collaborating authors while maintaining coherence and clarity. For this task, we directly retrieve the abstracts from the selected papers using the Semantic Scholar API [2]. The retrieved abstracts are then linked to their respective papers and author profiles.

**PSW-4: Paper Title Generation:** This task aims to generate a suitable title for the collaborative research paper, considering the user profiles, research interests, research questions,

---

[1]https://github.com/scholarly-python-package/scholarly

[2]https://api.semanticscholar.org/

and paper abstract. The generated title should be concise, informative, and reflect the paper's main contributions while considering the collaborating authors' expertise and interests. The data for this task is collected using the Semantic Scholar API, which provides the titles of the selected papers.

## 3.3 G-Eval for PSW Evaluation

We use the G-Eval framework [Liu *et al.*, 2023] to evaluate the generated outputs on the PSW benchmark. G-Eval employs LLMs like GPT-4 with chain-of-thought prompting to assess the quality of generated text in a form-filling paradigm [Zhang *et al.*, 2019a]. G-Eval is particularly well-suited for evaluating the PSW tasks because it can handle open-ended generation tasks without requiring gold reference outputs and provides scores that closely approximate expert human judgments [Yuan *et al.*, 2021]. We can use the G-Eval framework to obtain multi-dimensional evaluations of PSW model outputs. These dimensions include consistency, fluency, relevance, and novelty, which are considered essential scientific writing criteria [Kryściński *et al.*, 2019; Fabbri *et al.*, 2021]. An example G-Eval prompt can be found in Appendix C.

## 4 Experimental Setup

We assess our methods alongside other baseline approaches across the LaMP and PSW datasets. This section provides a detailed exploration of the experimental settings for these evaluations.

### 4.1 Datasets and Evaluation

**LaMP Dataset:** We follow the standard practice established in Salemi *et al.* [2023], encompassing three classification and four text generation tasks. Specifically, these tasks are Personalized Citation Identification (LaMP-1), Personalized News Categorization (LaMP-2), Personalized Product Rating (LaMP-3), Personalized News Headline Generation (LaMP-4), Personalized Scholarly Title Generation (LaMP-5), Personalized Email Subject Generation (LaMP-6), and Personalized Tweet Paraphrasing (LaMP-7).

**PSW Dataset:** As introduced in the previous section, the PSW dataset is designed to assess the performance of personalized language models in collaborative scientific writing scenarios. The dataset includes one individual task, User Profiling (UP-0), and four collaborative tasks: Research Topics Generation (PSW-1), Research Question Generation (PSW-2), Paper Abstract Generation (PSW-3), and Paper Title Generation (PSW-4).

**Evaluation:** Our evaluation methodology mirrors the LaMP framework outlined in Salemi *et al.* [2023]. We evaluate our proposed methods using the metrics specified in the LaMP benchmark for each task. These include F1 score, Accuracy, MAE, and RMSE for classification tasks and ROUGE-1 and ROUGE-L for generation tasks.

### 4.2 Methods to compare

We employ *gpt-3.5-turbo* hosted by OpenAI for all tasks in this paper. Our proposed method is compared against several baselines, including non-personalized language models, models fine-tuned on history data without personalization, and models that use simple concatenation of user histories for personalization.

## 4.3 Main Result

| Dataset | Metric | Non-personalized | | Personalized | | STEP-BACK PROFILING |
|---------|--------|---------|--------|----------|--------|---------------------|
| | | FlanT5-XXL | ChatGPT | FlanT5-XXL | ChatGPT | |
| LaMP-1 | Accuracy | 0.522 | 0.510 | 0.675 | **0.701** | 0.624 |
| LaMP-2 | Accuracy | 0.591 | 0.610 | 0.598 | 0.693 | **0.729** |
| | F1 | 0.463 | 0.455 | 0.477 | 0.455 | **0.591** |
| LaMP-3 | MAE | 0.357 | 0.699 | 0.282 | 0.658 | **0.274** |
| | RMSE | 0.666 | 0.977 | 0.584 | 1.102 | **0.559** |
| LaMP-4 | ROUGE-1 | 0.164 | 0.133 | 0.192 | 0.160 | **0.195** |
| | ROUGE-L | 0.149 | 0.118 | 0.178 | 0.142 | **0.180** |
| LaMP-5 | ROUGE-1 | 0.455 | 0.395 | 0.467 | 0.398 | **0.469** |
| | ROUGE-L | 0.410 | 0.334 | 0.424 | 0.336 | **0.426** |
| LaMP-6 | ROUGE-1 | 0.332 | - | 0.466 | - | **0.485** |
| | ROUGE-L | 0.320 | - | 0.453 | - | **0.464** |
| LaMP-7 | ROUGE-1 | **0.459** | 0.396 | 0.448 | 0.391 | 0.455 |
| | ROUGE-L | **0.404** | 0.337 | 0.396 | 0.324 | 0.398 |

Table 1: Performance comparison of personalized and non-personalized models on the LaMP dataset.

**LaMP Results:** To guarantee a fair comparison, we utilized a user-based separation from Salemi *et al.* [2023]. We only granted the agent access to the provided user history and restricted it from accessing any other information. Additionally, we utilized the same pre-trained retriever, without any additional fine-tuning, to retrieve the top five examples. This approach was identical to the *Non-Personalized* setting in Salemi *et al.* [2023]. Finally, we compared our results with the outcomes reported in the study.

As shown in Table 1[3], our analysis unveiled a notable performance enhancement through our method's application, significantly when leveraging the same backbone language models (*gpt-3.5-turbo*). In the domain of text generation tasks (LaMP-4~7), our method achieved an average improvement of 0.048 in Rouge-1 and 0.053 in Rouge-L, corresponding to gains of 15.2% and 19.5%, respectively. Similarly, for the classification tasks (LaMP-1~3), we observed an average +12.6% accuracy gain of and a +42.5% reduction in MAE compared to the *Non-Personalized* setting. Our method continues to exhibit better performance across most tasks, even when compared with **FlanT5-XXL**, with a fine-tuned retriever as *Personalized* setting. The prompt used in this experiment is detailed in Appendix D.

**PSW Results**

We assess the proficiency of our proposed personalized agent using the PSW dataset, focusing on user profiling (UP-0), personalized idea brainstorming (PSW-1, PSW-2), and personalized text generation (PSW-3, PSW-4). We compare the performance of our method in three different settings:

1. **Zero-shot**: Generates outputs based on the input prompt $x$ alone:

$$y = \text{Generate}(x).$$

---

[3]Baseline results are obtained directly from Salemi *et al.* [2023].

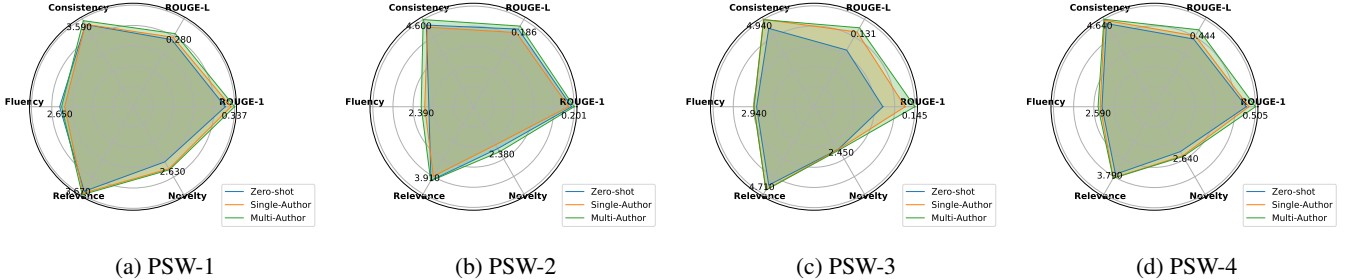

| | (a) PSW-1 | (b) PSW-2 | (c) PSW-3 | (d) PSW-4 |

Figure 3: PSW datasets' Performance metrics (ROUGE-1, ROUGE-L, Consistency, Fluency, Relevance, and Novelty) across three different models: **Zero-shot**, **Single-Author**, and **Multi-Author**. The **Multi-Author** model consistently achieves the highest scores across all datasets.

| Datasets | Method | Metrics | | | | | |
|---|---|---|---|---|---|---|---|
| | | ROUGE-1 | ROUGE-L | Consistency | Fluency | Relevance | Novelty |
| UP-0 | Single-Author | 0.267 | 0.233 | 4.32 | 2.01 | 3.59 | / |
| PSW-1 | Zero-shot | 0.306 | 0.257 | 3.43 | **2.65** | 3.53 | 2.30 |
| | Single-Author | 0.325 | 0.266 | 3.44 | 2.47 | 3.61 | 2.59 |
| | Multi-Author | **0.337** | **0.280** | **3.59** | 2.58 | **3.67** | **2.63** |
| PSW-2 | Zero-shot | 0.196 | 0.179 | 4.31 | 2.04 | 3.89 | 2.21 |
| | Single-Author | 0.190 | 0.171 | 4.20 | 2.23 | 3.67 | 2.01 |
| | Multi-Author | **0.201** | **0.186** | **4.60** | **2.39** | **3.91** | **2.38** |
| PSW-3 | Zero-shot | 0.099 | 0.094 | 4.43 | 2.81 | 4.43 | 2.40 |
| | Single-Author | 0.131 | 0.124 | **4.94** | **2.94** | 4.70 | 2.40 |
| | Multi-Author | **0.145** | **0.131** | 4.92 | **2.94** | **4.71** | **2.45** |
| PSW-4 | Zero-shot | 0.459 | 0.391 | 4.41 | 2.41 | 3.58 | 2.38 |
| | Single-Author | 0.472 | 0.409 | 4.59 | 2.49 | 3.78 | 2.60 |
| | Multi-Author | **0.505** | **0.444** | **4.64** | **2.59** | **3.79** | **2.64** |

Table 2: Performance comparison of personalized models on the PSW dataset, with additional metrics such as **Consistency (1-5)**, **Fluency (1-3)**, **Relevance (1-5)**, and **Novelty (1-3)** reported.

2. **Single-Author**: Personalizes with single user's profile $P_{u_i}$ and retrieved snippets $R_i$:

$$y = \text{Generate}(\hat{x}, P_{u_i}),$$

where $\hat{x} = [x; R_i]$ and $R_i = \text{Retrieve}(x, H_i, 10)$.

3. **Multi-Author**: Personalizes with multiple users' profiles $P_U$ and retrieved snippets $R$:

$$y = \text{Generate}(\hat{x}, P_U),$$

where $\hat{x} = [x; R_1; \cdots; R_n]$, $R_i = \text{Retrieve}(x, H_i, 10)$ for each user $u_i$.

As shown in Table 2, our **Multi-Author** setting demonstrates superior performance across all tasks. In the personalized idea brainstorming tasks (PSW-1 and PSW-2), the **Multi-Author** setting outperforms both **Zero-shot** and **Single-Author** settings, with an average improvement of +6.9% in ROUGE-1 and +7.1% in ROUGE-L. Similarly, for the personalized text generation tasks (PSW-3 and PSW-4), the **Multi-Author** setting achieves the highest ROUGE scores, with an average gain of +28.2% in ROUGE-1 and +26.6% in ROUGE-L, compared to the **Zero-shot** and **Single-Author** settings. Furthermore, the **Multi-Author** setting exhibits the highest scores for additional metrics such as Consistency, Fluency, Relevance, and Novelty across all tasks, with an average improvement of

+5.1%, +6.7%, +3.8%, and +6.4%, respectively, compared to the **Zero-shot** and **Single-Author** setting. The prompt used in this experiment is detailed in Appendix E.

## 4.4 Ablation Studies

To assess the contribution of each component, we perform an ablation study on the PSW dataset. Table 3 and 4 report the results of two variants: 1) Switching the order of users and 2) Removing user profiling.

**Impact of Author Order**

Table 3 shows how changing the author order affects the performance of multi-user personalized models. We experiment with three variants:

- **Original**: The original author order as provided in the dataset.
- **Swap-Random**: Randomly shuffle the order of authors.
- **Swap-First**: Move the first author to the end of the author list.

| Datasets | Variants | Metrics | | | | | |
|---|---|---|---|---|---|---|---|
| | | ROUGE-1 | ROUGE-L | Consistency | Fluency | Relevance | Novelty |
| PSW-1 | Original | **0.337** | **0.280** | **3.59** | **2.58** | 3.67 | **2.63** |
| | Swap-Random | 0.321 | 0.272 | 3.42 | 2.48 | **3.69** | 2.45 |
| | Swap-First | 0.314 | 0.260 | 3.35 | 2.42 | 3.48 | 2.37 |
| PSW-2 | Original | **0.201** | **0.186** | **4.60** | **2.39** | **3.91** | 2.38 |
| | Swap-Random | 0.193 | 0.178 | 4.53 | 2.30 | 3.85 | **2.42** |
| | Swap-First | 0.186 | 0.171 | 4.46 | 2.27 | 3.77 | 2.29 |
| PSW-3 | Original | **0.145** | **0.131** | **4.92** | 2.94 | **4.71** | 2.45 |
| | Swap-Random | 0.138 | 0.125 | 4.84 | 2.88 | 4.65 | 2.50 |
| | Swap-First | 0.130 | 0.117 | 4.78 | **2.98** | 4.57 | 2.55 |
| PSW-4 | Original | **0.505** | **0.444** | **4.64** | **2.59** | **3.79** | 2.64 |
| | Swap-Random | 0.492 | 0.431 | 4.57 | 2.55 | 3.72 | 2.70 |
| | Swap-First | 0.483 | 0.421 | 4.50 | 2.50 | 3.64 | **2.76** |

Table 3: Impact of author order on the performance of multi-user personalized models, with additional metrics such as **Consistency (1-5)**, **Fluency (1-3)**, **Relevance (1-5)**, and **Novelty (1-3)** reported.

The **Original** order consistently achieves the best performance across all metrics on all PSW tasks. Randomly swapping authors (**Swap-Random**) leads to a slight decline, while moving the first author to the end (**Swap-First**) results in a more significant drop. This observation highlights the importance of preserving the original author order

in multi-author collaborative writing scenarios. The first author, often the lead or corresponding author, significantly influences the document's content, structure, and style. As a result, their writing style and expertise tend to be most prominently reflected in the document. Disrupting this order introduces noise and hinders the model's ability to capture the individual authors' impact and the logical progression of ideas, particularly affecting the generation tasks (PSW-3 and PSW-4) where content and style are heavily influenced by the main author's expertise and preferences.

**Impact of User Profiling**

Table 4 reports ablation results on the user profile component:

- **Original**: User profiles constructed using STEP-BACK PROFILING.

- **Removed**: No user profiles used, only retrieving relevant snippets.

- **Random**: Replacing target user profiles with randomly sampled user profiles.

| Datasets | Profile | Metrics | | | | | |
|---|---|---|---|---|---|---|---|
| | | ROUGE-1 | ROUGE-L | Consistency | Fluency | Relevance | Novelty |
| PSW-1 | Original | **0.337** | **0.280** | **3.59** | **2.58** | **3.67** | 2.63 |
| | Removed | 0.297 | 0.250 | 3.21 | 2.49 | 3.31 | 2.57 |
| | Random | 0.328 | 0.272 | 3.55 | 2.56 | 3.62 | **2.68** |
| PSW-2 | Original | **0.201** | **0.186** | **4.60** | 2.39 | **3.91** | 2.38 |
| | Removed | 0.180 | 0.166 | 4.28 | 2.32 | 3.63 | 2.33 |
| | Random | 0.195 | 0.182 | 4.57 | **2.42** | 3.89 | **2.45** |
| PSW-3 | Original | **0.145** | **0.131** | 4.92 | 2.94 | **4.71** | 2.45 |
| | Removed | 0.128 | 0.115 | 4.70 | 2.87 | 4.50 | 2.41 |
| | Random | 0.142 | 0.128 | **4.95** | **2.96** | 4.69 | **2.51** |
| PSW-4 | Original | **0.505** | **0.444** | **4.64** | **2.59** | **3.79** | 2.64 |
| | Removed | 0.475 | 0.419 | 4.38 | 2.53 | 3.58 | 2.56 |
| | Random | 0.498 | 0.438 | 4.60 | 2.58 | 3.76 | **2.69** |

Table 4: Impact of user profile on the performance of multi-user personalized models, with additional metrics such as **Consistency (1-5)**, **Fluency (1-3)**, **Relevance (1-5)**, and **Novelty (1-3)** reported.

Removing user profiles (**Removed**) leads to the largest performance decline, confirming the benefit of STEP-BACK PROFILING in multi-user personalization. Using random profile texts (**Random**) recovers some of the gaps but still underperforms the **Original** profiles. This demonstrates that the distilled user traits successfully capture useful information for collaborative writing, such as individual writing styles, expertise, and preferences. The performance gap between **Original** and **Random** profiles highlights the effectiveness of the STEP-BACK PROFILING technique in extracting relevant user characteristics from their background information. These findings underscore the importance of incorporating author-specific traits to enable a more personalized and contextually appropriate generation in multi-user settings.

## 5 Conclusion

In summary, STEP-BACK PROFILING offers a promising way to improve the effectiveness and scalability of personalized language models. Abstracting user histories into compact profiles enables the model to better focus on pertinent information and handle longer contexts. Experiments on both single-user and multi-user settings validate the benefits of profile-guided personalization.

Future work can explore more advanced profiling strategies, such as hierarchical representations and dynamic profile updates based on user feedback. Adapting STEP-BACK PROFILING to long histories spanning multiple sessions is another valuable direction. Finally, studying the interpretability and controllability of profile-guided models can help build user trust and allow for more fine-grained customization.

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

# C  Details of G-Eval Prompt

| **Task Description** |
| --- |
| You will be given one result generated for a science paper and several reference papers. Your task is to rate the result using the following criteria. 
 Please make sure you read and understand these instructions carefully. Please keep this document open while reviewing, and refer to it as needed. |
| **Evaluation Criteria** |
| **Consistency (1-5)** – the factual alignment between the result and the corresponding science paper. A factually consistent result contains only statements entailed by the source document. 
 **Fluency (1-3)** – the quality of the result in terms of grammar, spelling, punctuation, word choice, and sentence structure. 
 **Relevance (1-5)** – the selection of important content from the source. The result should include only important information from the source document. 
 **Novelty (1-3)** – the uniqueness and originality of the result in terms of concept, perspective, and creativity. |
| **Evaluation Task** |
| Now, you are working on evaluating this prediction: 
 {Prediction Text} 
 Here are some ground truth results for comparison: [$result_1$, $result_2$, . . . ]. |
| **Instruction** |
| Please evaluate the prediction using the above criteria. |

Table 6: Prompt template for evaluating the G-Eval metric.

# D  Prompts for LaMP Tasks

## D.1  Personalized Citation Identification (LaMP-1) Prompt

| **User Profile** |
| --- |
| Assuming you care a lot about these areas: 
 **Keywords:** [$keyword_1$, $keyword_2$, $keyword_3$, . . . ] 
 **Topics:** [$topics_1$, $topics_2$, $topics_3$, . . . ] |
| **User History** |
| I give you some titles of papers that you've written. Please imitate your reasons and recommend a paper citation for me. Each example consists of an abstract, the corresponding title, and a description of the writing style and keywords for that title. |
| **Example 1** |
| **Title:** {Title Text} 
 **Abstract:** {Abstract Text} 
 **Reason:** {Reason} 
 **Citation:** [$citation_1$, $citation_2$, . . . ] |
| **Example 2** |
| **Title:** {Title Text} 
 **Abstract:** {Abstract Text} 
 **Reason:** {Reason} 
 **Citation:** [$citation_1$, $citation_2$, . . . ] 
 . . . |
| **Example k** |
| **Title:** {Title Text} 
 **Abstract:** {Abstract Text} 
 **Reason:** {Reason} 
 **Citation:** [$citation_1$, $citation_2$, . . . ] |
| **Classification Task** |
| Now you have written this title: 
 **Title:** {Title Text} |
| **Instruction** |
| Please separately analyze the potential relevant connection of **Reference 1** and **Reference 2** to this title. You are citing from one of them. Please decide which one it would be: 
 **Reference 1:** {$option_1$} 
 **Reference 2:** {$option_2$} 
 Just answer with [1] or [2] without explanation. |

Table 7: Prompt template for the Personalized Citation Identification (LaMP-1) task.

## D.2 Personalized News Categorization (LaMP-2) Prompt

| **User Profile** |
| --- |
| Assuming you care a lot about these areas: **Keywords:** [keyword$_1$, keyword$_2$, keyword$_3$, ...] **Topics:** [topics$_1$, topics$_2$, topics$_3$, ...] |
| **User History** |
| I give you some titles and articles that you've written with category. Please imitate your reasons for giving this category. Each example consists of an abstract, the corresponding title, and a category of it. |
| **Example 1** |
| **Article:** {Article Text} **Title:** {Title Text} **Reason:** {Reason} **Category:** [category$_1$, category$_2$, ...] |
| **Example 2** |
| **Article:** {Article Text} **Title:** {Title Text} **Reason:** {Reason} **Category:** [category$_1$, category$_2$, ...] ... |
| **Example k** |
| **Article:** {Article Text} **Title:** {Title Text} **Reason:** {Reason} **Category:** [category$_1$, category$_2$, ...] |
| **Classification Task** |
| Now you have written this article with the title: **Article:** {Article Text} **Title:** {Title Text} |
| **Instruction** |
| Which category does this article relate to among the following categories? **Category 1:** {option$_1$} **Category 2:** {option$_2$} ... **Category K:** {option$_N$} Just answer with the category name without further explanation. |

Table 8: Prompt template for the Personalized News Categorization (LaMP-2) task.

## D.3 Personalized Product Rating (LaMP-3) Prompt

| **User Profile** |
| --- |
| Assuming you have written product reviews with the following characteristics: **Most Common Rating:** {score$_{most}$} **Rating Patterns:** [pattern$_1$, pattern$_2$, ...] |
| **User History** |
| I provide you with some product reviews you've written, along with their corresponding ratings. Please imitate your reasoning for assigning these ratings. Each example consists of a product review and its rating. |
| **Example 1** |
| **Product Review:** {Review Text} **Rating:** {Rating} |
| **Example 2** |
| **Product Review:** {Review Text} **Rating:** {Rating} ... |
| **Example k** |
| **Product Review:** {Review Text} **Rating:** {Rating} |
| **Rating Task** |
| Now you have written this new product review: **Product Review:** {Review Text} Based on the review, please analyze its sentiment and how much you like the product. |
| **Instruction** |
| Follow your previous rating habits and these instructions: |
| • If you feel satisfied with this product or have concerns but it's good overall, it should be rated 5. • If you feel good about this product but notice some issues, it should be rated as 4. • If you feel OK but have concerns, it should be rated as 3. • If you feel unsatisfied with this product but it's acceptable for some reason, it should be rated as 2. • If you feel completely disappointed or upset, it should be rated 1. |
| Your most common rating is {score$_{most}$}. You must follow this rating pattern faithfully and answer with the rating without further explanation. |

Table 9: Prompt template for the Personalized Product Review Rating (LaMP-3) task.

## D.4 Personalized News Headline Generation (LaMP-4) Prompt

| **User Profile** |
| --- |
| Assuming you have written headlines with the following characteristics:
**Writing Style:** [style$_1$, style$_2$, . . . ]
**Content Patterns:** [patterns$_1$, patterns$_2$, . . . ] |
| **User History** |
| I will provide you with some news articles along with the headlines you've written for them. Please imitate your writing style and content patterns when generating a new headline. Each example consists of a news article and its corresponding headline. |
| **Example 1** |
| **Article:** {Article Text}
**Headline:** {Headline} |
| **Example 2** |
| **Article:** {Article Text}
**Headline:** {Headline}
. . . |
| **Example k** |
| **Article:** {Article Text}
**Headline:** {Headline} |
| **Generation Task** |
| Now that you have been given this news article:
**Article:** {Article Text} |
| **Instruction** |
| Please write a headline following your previous writing styles and habits. If you have written headlines with similar content, you could reuse those headlines and mimic their content. |

Table 10: Prompt template for the Personalized News Headline Generation (LaMP-4) task.

## D.5 Personalized Scholarly Title Generation (LaMP-5) Prompt

| **User Profile** |
| --- |
| Assuming you have written scholarly titles with the following characteristics:
**Writing Style:** [style$_1$, style$_2$, . . . ]
**Title Patterns:** [pattern$_1$, pattern$_2$, . . . ] |
| **User History** |
| I will provide you with some research paper abstracts along with the titles you've written for them. Please imitate your writing style and title patterns when generating a new title. Each example consists of a paper abstract and its corresponding title. |
| **Example 1** |
| **Abstract:** {Abstract Text}
**Title:** {Title} |
| **Example 2** |
| **Abstract:** {Abstract Text}
**Title:** {Title}
. . . |
| **Example k** |
| **Abstract:** {Abstract Text}
**Title:** {Title} |
| **Generation Task** |
| Now that you have been given this paper abstract:
**Abstract:** {Abstract Text} |
| **Instruction** |
| Please write a title following your previous style and habits, keeping it clear, accurate, and concise. |

Table 11: Prompt template for the Personalized Scholarly Title Generation (LaMP-5) task.

## D.6 Personalized Email Subject Generation (LaMP-6) Prompt

| |
|---|
| **User Profile** |
| Assuming you care a lot about these areas: 
 **Keywords:** [keyword$_1$, keyword$_2$, keyword$_3$, . . . ] 
 **Topics:** [topics$_1$, topics$_2$, topics$_3$, . . . ] |
| **User History** |
| Let's say there are some emails you've written. Please mimic the style of these examples. Each example consists of email content, the corresponding subject, and a description of the writing style for that title. |
| **Example 1** |
| **Content:** {Email Content} 
 **Writing Style:** {Style} 
 **Subject:** {Email Subject} |
| **Example 2** |
| **Content:** {Email Content} 
 **Writing Style:** {Style} 
 **Subject:** {Email Subject} 
 . . . |
| **Example k** |
| **Content**: {Email Content} 
 **Writing Style:** {Style} 
 **Subject**: {Email Subject} |
| **Generation Task** |
| Now that you have been given this email content: 
 **Content:** {Email Content} |
| **Instruction** |
| Write a title following your previous style and habits. Just answer with the subject without further explanation. |

Table 12: Prompt template for the Personalized Email Subject Generation (LaMP-6) task.

## D.7 Personalized Tweet Paraphrasing (LaMP-7) Prompt

| |
|---|
| **User Profile** |
| Assuming you have written tweets with the following characteristics: 
 **Writing Style:** [style$_1$, style$_2$, . . . ] 
 **Tone:** [tone$_1$, tone$_2$, . . . ] 
 **Length:** [length$_1$, length$_2$, . . . ] |
| **User History** |
| I will provide you with some original tweets along with the paraphrased versions you've written for them. When paraphrasing a new tweet, please imitate your writing style, tone, and typical length. Each example consists of an original tweet and its paraphrased version. |
| **Example 1** |
| **Original Tweet:** {Tweet Text} 
 **Paraphrased Tweet:** {Paraphrased Text} |
| **Example 2** |
| **Original Tweet:** {Tweet Text} 
 **Paraphrased Tweet:** {Paraphrased Text} 
 . . . |
| **Example k** |
| **Original Tweet:** {Tweet Text} 
 **Paraphrased Tweet:** {Paraphrased Text} |
| **Generation Task** |
| Now that you have been given this tweet: 
 **Original Tweet:** {Tweet Text} |
| **Instruction** |
| Please paraphrase it with the following instructions: 
 • You must use tweet styles and tones. 
 • You must keep it faithful to the given tweet with similar keywords and length. |

Table 13: Prompt template for the Personalized Tweet Paraphrasing (LaMP-7) task.

# E Prompts for PSW Tasks

## E.1 Research Interests Generation (UP-0) Prompt

| **User History** |
| --- |
| I will provide you with some research papers you've authored. Please summarize your top research interests based on these papers. Each paper consists of a title and abstract. |
| **Paper 1** |
| **Title:** {Title Text} 
 **Abstract:** {Abstract Text} |
| **Paper 2** |
| **Title:** {Title Text} 
 **Abstract:** {Abstract Text} |
| . . . |
| **Paper k** |
| **Title:** {Title Text} 
 **Abstract:** {Abstract Text} |
| **Instruction** |
| Please summarize your top three research interests based on the provided papers in the following format: 
 **Research Interests:** [interest$_1$, interest$_2$, interest$_3$, . . . ] |

Table 14: Prompt template for the Research Interests Generation (UP-0) task.

## E.2 Personalized Research Paper Title Generation (PSW-1) Prompt

| **User Profile** |
| --- |
| Assuming you are an expert researcher with the following research interests: 
 **Research Interests:** [interest$_1$, interest$_2$, interest$_3$, . . . ] |
| **User History** |
| Here are some titles and abstracts from papers you have authored: |
| **Paper 1** |
| **Title:** {Title} 
 **Abstract:** {Abstract} |
| **Paper 2** |
| **Title:** {Title} 
 **Abstract:** {Abstract} |
| . . . |
| **Paper k** |
| **Title:** {Title} 
 **Abstract:** {Abstract} |
| **Brainstorm Task** |
| Here are some related papers for reference, each with a title: 
 **Reference 1:** {Title} 
 **Reference 2:** {Title} 
 . . . 
 **Reference N:** {Title} |
| **Instruction** |
| Considering your research interests, previous works, and reference papers, please brainstorm the most promising title for your new research paper. |

Table 15: Prompt template for the Personalized Research Paper Title Generation (PSW-1) task.

## E.3 Research Question Generation (PSW-2) Prompt

**User Profile**

Assuming you are an expert researcher with the following research interests:

**Research Interests:** [$interest_1$, $interest_2$, $interest_3$, ...]

**User History**

Here are some titles and abstracts from papers you have authored:

**Paper 1**

**Title:** {Title}

**Abstract:** {Abstract}

**Paper 2**

**Title:** {Title}

**Abstract:** {Abstract}

...

**Paper k**

**Title:** {Title}

**Abstract:** {Abstract}

**Brainstorm Task**

Now you are working on a new paper with the following title:

**Title:** {Title}

**Instruction**

Considering the title and research background, please propose the top 3 research questions you aim to address in this new paper.

Table 16: Prompt template for the Research Question Generation (PSW-2) task.

## E.4 Paper Abstract Generation (PSW-3) Prompt

**User Profile**

Assuming you are an expert researcher with the following research interests:

**Research Interests:** [$interest_1$, $interest_2$, $interest_3$, ...]

**User History**

Here are some titles and abstracts from papers you have authored:

**Paper 1**

**Title:** {Title}

**Abstract:** {Abstract}

**Paper 2**

**Title:** {Title}

**Abstract:** {Abstract}

...

**Paper k**

**Title:** {Title}

**Abstract:** {Abstract}

**Generation Task**

Now you are working on a new paper with the following title:

**Title:** {Title}

And you are focusing on solving the following research questions: [$question_1$, $question_2$, ...]

**Instruction**

Considering the title, research questions, and your writing style in previous abstracts, please write an abstract for this new paper.

Table 17: Prompt template for the Paper Abstract Generation (PSW-3) task.

 ## E.5 Paper Title Generation (PSW-4) Prompt

| |
|---|
| **User Profile** |
| Assuming you are an expert researcher with the following research interests: |
| **Research Interests:** [$interest_1$, $interest_2$, $interest_3$, ...] |
| **User History** |
| Here are some titles and abstracts from papers you have authored: |
| **Paper 1** |
| **Title:** {Title} |
| **Abstract:** {Abstract} |
| **Paper 2** |
| **Title:** {Title} |
| **Abstract:** {Abstract} |
| ... |
| **Paper k** |
| **Title:** {Title} |
| **Abstract:** {Abstract} |
| **Generation Task** |
| Now, you are working on a new paper with the following abstract: |
| **Abstract:** {Abstract} |
| And you are focusing on solving the following research questions: [$question_1$, $question_2$, . . . ] |
| **Instruction** |
| Considering the abstract and your title writing style in previous papers, please generate a title for this new paper. The title should be clear and concise and reflect the main topic of the abstract as well as your research questions. |

Table 18: Prompt template for the Paper Title Generation (PSW-4) task.