# OpenReview forum: "Step-Back Profiling: Distilling User Interactions for Personalized Scientific Writing"
_ijcai.org/IJCAI/2024/Workshop/AI4Research — AI4Research 2024_

### Official Review · Reviewer_sVBp · 2024-05-31
**A well-written paper about personalizes large language models for scientific writing by utilizing multi-user profiles.**

**Rating:** 7
**Confidence:** 3

**Review:**

The idea is well-motivated, and the method description is clear, but experiments could be improved.

Pros:

- This paper introduces a training-free technique, STEP-BACK PROFILING, for personalizing large language models by distilling user interactions using gist into concise profiles.

- The authors extend the LaMP dataset into the Personalized Scientific Writing (PSW) dataset to evaluate multi-user scenarios in collaborative scientific writing.

- The proposed method is effective on the LaMP and PSW datasets.

Cons:

- It is unclear whether "gist"-style information compression is necessary or if another retrieval method would be more effective, as no comparisons or ablation studies have been conducted.

- Although the toolkit provided by LaMP for evaluating its generative tasks utilizes ROUGE-1 and ROUGE-L metrics, employing more advanced methods like BERTScore and large language models (LLMs) is preferable as they better understand the semantic meaning of the generated text rather than just the n-gram overlap. For instance, [1] employs GPT-4 to evaluate generative tasks using its proposed method on the LaMP dataset.

---

[1] Zhang, Kaiyan, et al. "CoGenesis: A Framework Collaborating Large and Small Language Models for Secure Context-Aware Instruction Following." arXiv preprint arXiv:2403.03129 (2024).

---

### Official Review · Reviewer_uDvM · 2024-06-04
**STEP-BACK PROFILING**

**Rating:** 8
**Confidence:** 3

**Review:**

The paper introduces STEP-BACK PROFILING, a novel approach to personalizing large language models (LLMs) by condensing user interaction histories into concise, trait-centric profiles. This technique aims to improve the generation of personalized content, specifically in the context of scientific writing. The main problem addressed is the challenge LLMs face in creating individualised content that effectively captures a user's unique style and preferences, especially in multi-user or collaborative scenarios. The paper is well-written and easy to follow. The experiments and analysis are also solid.

Strengths:
A training-free profiling method, the task itself seems interesting and is relevant to the conference.
Enhanced personalisation: Improves content generation without extra training.
The proposed benchmark should be useful for future research.

Weaknesses:
I am not an expert in this area, but this task setting seems a little bit unrealistic to me. I guess the proposed method might have some challenges in dynamically integrating diverse user traits, especially since the author has investigated the impact of different user orders.

---

### Decision · Program_Chairs · 2024-06-03

Accept